# A Glimpse into the Role and Effectiveness of Splenectomy for Isolated Metachronous Spleen Metastasis of Colorectal Cancer Origin: Long-Term Survivals Can Be Achieved

**DOI:** 10.3390/jcm13082362

**Published:** 2024-04-18

**Authors:** Beatrice Mihaela Tivadar, Traian Dumitrascu, Catalin Vasilescu

**Affiliations:** Department of General Surgery, Fundeni Clinical Institute, Carol Davila University of Medicine and Pharmacy, Fundeni Street No. 258, 022328 Bucharest, Romania; mihaela-beatrice.tivadar@drd.umfcd.ro (B.M.T.); catalin.vasilescu@umfcd.ro (C.V.)

**Keywords:** spleen metastasis, colorectal cancer, splenectomy, survival

## Abstract

**Background**: Many papers exploring the role of resectioning metastases in colorectal cancer (CRC) have focused mainly on liver and lung sites, showing improved survival compared with non-resectional therapies. However, data about exceptional metastatic sites such as splenic metastases (SMs) are scarce. This paper aims to assess the role and effectiveness of splenectomy in the case of isolated metachronous SM of CRC origin. **Methods**: The patients’ data were extracted after a comprehensive literature search through public databases for articles reporting patients with splenectomies for isolated metachronous SM of CRC origin. Potential predictors of survival were explored, along with demographic, diagnostic, pathology, and treatment data for each patient. **Results**: A total of 83 patients with splenectomies for isolated metachronous SM of CRC origin were identified. The primary CRC was at an advanced stage (Duke’s C—70.3%) and on the left colon (45.5%) for most patients, while the median interval between CRC resection and SM was 24 months. The median overall survival after splenectomy was 84 months, and patients younger than 62 years presented statistically significantly worse overall survival rates than those ≥62 years old (*p* = 0.011). There was no significant impact on the long-term outcomes for factors including primary tumor location or adjuvant chemotherapy (*p* values ≥ 0.070, ns). Laparoscopic splenectomy was increasingly used in the last 20 years from 2002 (33.3% vs. 0%, *p* < 0.001). **Conclusions**: Splenectomy is the optimal treatment for patients with isolated metachronous SM of CRC, with the laparoscopic approach being increasingly used and having the potential to become a standard of care. Encouraging long-term survival rates were reported in the context of a multidisciplinary approach. Younger ages are associated with worse survival. Perioperative chemotherapy in the context of a patient diagnosed with SM of CRC origin appears to be a reasonable option, although the present study failed to show any significant impact on long-term survival.

## 1. Introduction

Colorectal cancer (CRC) is a common cancer worldwide, being estimated in 2024 to be the third most frequent type of cancer, both in men and women, in the United States. Although over time, the CRC death rate has decreased in both men and women, CRC is still estimated to represent the third cause of death by cancer in men and the fourth cause of death by cancer in women [1]. Due to population growth and demographic aging, the number of new CRC patients is expected to increase in many countries, while the treatment costs with CRC patients are substantial [2].

Approximately 22% of patients with CRC have metastatic disease at diagnosis, while 38% of patients have regional disease, and 32% of patients have localized disease. The stage of the disease at the time of diagnosis strongly influences the prognosis of a patient with CRC. Thus, the five-year survival of CRC patients with distant metastasis is only 14%, while for the regional and localized diseases, the five-year survival is 73% and 91%, respectively [1]. Geographical disparities regarding the survival rates were observed for patients diagnosed with CRC. Thus, better survival rates were observed in Western European countries compared with Central and South American countries, a situation explained by better screening programs and treatment options in Western Europe [2].

In the United States, in the last few years, there has been a trend toward increasing incidence of CRC in individuals younger than 50 years. The prevalence of advanced CRC is higher in younger versus older patients, a situation explained by the lack of screening in asymptomatic younger patients and higher percentages of misdiagnosis in younger symptomatic patients compared with the older ones [2,3].

Nowadays, managing a patient with CRC is multimodal, and each patient’s situation should be discussed in a multidisciplinary tumor board to maximize the patient’s chances of survival. Higher survival rates can be obtained even for advanced CRC with developments in surgery, chemotherapy, and radiotherapy according to cancer location, stage, and patient performance status.

Distant metastases in CRC can be present at the time of diagnosis (synchronous metastases) or occur at an interval after resection of the primary tumor (metachronous metastases). A few studies identified synchronous metastases at diagnosis of CRC in 17.7% to 22% of patients, while metachronous metastases occurred in 18% to 70% of patients after resection of the primary tumor [4,5,6,7,8,9].

For metastatic CRC, there is an improvement in survival for patients with a multimodal approach, where surgery (i.e., resection) plays a crucial role [6,9,10]. Furthermore, the indication for resection is extended in metastatic CRC, particularly for those patients with single-site metastasis [8]. Thus, significant improvements in survival have been observed, particularly for liver and lung metastases of CRC, with the advances in surgical methods, cancer-directed therapy, chemo-radiotherapy, and optimization of targeted therapies [1,6,11,12].

The most frequent pattern of metastatic disease in CRC is represented by the liver and/or lung (up to 91.% of patients with synchronous metastases and up to 81.5% of patients with metachronous metastases) [5,7,8,9,13]. A Dutch study including 5671 CRC patients with curative intent surgery has identified as the most common sites of metachronous metastatic disease the liver (60%), lungs (39%), extra-regional lymph nodes (22%), and peritoneum (19%), other less frequent sites being the brain and the bones [4]. Another Dutch study, including 160,278 patients diagnosed with CRC, identified synchronous metastases in 21% of the patients, with liver, lung, and peritoneum being the most frequent sites [6]. Other isolated sites of metastatic diseases in CRC are rare. Resection of metastases of CRC origin has been associated with significant improvement in survival compared with that of patients who did not undergo resection, particularly for liver, lung, and even limited peritoneal metastases [6,8,14,15].

Metastases in CRC have multiple sites in many patients for synchronous and metachronous settings [4,6,9,13]. Isolated metachronous metastases of CRC are rare, the most frequent sites being the liver (29%), the lung (10%), and the peritoneum (8%); isolated single-site metastases to other organs represent less than 5% of patients [4].

Many papers exploring the role and effectiveness of resection of metastases in CRC have focused mainly on liver and lung sites or limited peritoneal disease [8]. Only a few papers have examined the potential role of resection for rare sites of CRC metastases, such as splenic metastases (SMs) [16,17,18,19,20,21,22]. Thus, SM from CRC generally occurs in the context of advanced metastatic disease, usually after cancer has spread to sites such as the liver, lungs, peritoneum, and other sites [4,7,23]. Diagnosing isolated SM of CRC without further organ involvement is exceptional, representing less than 0.2% of resected metastases of CRC origin [18,24].

While the reason metastases of CRC rarely occur in the spleen is not fully understood, the question remains: can these patients with isolated SM of CRC benefit from surgical resection or not? Currently, the diagnosis and long-term benefit of resection for isolated SM of CRC origin are poorly addressed in case report papers and a few literature reviews, including a limited number of patients. The present study aims to assess the role and effectiveness of resection in isolated metachronous SM of CRC origin based on data provided by the current literature.

## 2. Patients and Methods

### 2.1. Patients

The patients’ data were extracted after a comprehensive literature search of the PubMed-Medline, Google Scholar, and Elsevier-Scopus databases for articles reporting data of patients with metachronous SM of CRC origin. The search strategy combined keywords such as “spleen metastasis”, “isolated spleen metastasis”, “colon cancer”, “rectal cancer”, “colorectal cancer”, and “splenectomy”. Up to January 2024, several publications were retrieved, and the search was refined, including only cases where there were no other metastatic sites at the time of diagnosis of SM and only the cases where there was a single-site metachronous SM of CRC origin, all treated by resection (i.e., splenectomy). Furthermore, the articles’ references were reviewed to identify potential other patients to be included in the analyses. No language restrictions were made. BT and TD independently screened the literature and extracted data; any disagreement was discussed and managed under CV’s supervision.

A few studies were excluded from the analysis if clear parameters of interest could not be extracted from the text or if essential results were not reported. From each study, we pulled data regarding the number, age, and gender of patients, the site of the primary tumor, the stage of the disease at the time of primary tumor resection, the interval between the primary tumor resection and the resection of SM, clinical signs and symptoms, imaging methods and carcinoembryonic antigen (CEA) serum level at the time of SM diagnosis, the number and size of metastases in the spleen, the type of resection/approach performed for the SM, data about oncological medical treatment, and lastly, the overall survival status of these patients.

All included articles described metachronous SM of CRC origin (adenocarcinoma-only patients) confirmed by pathology of the resected specimens, presenting as isolated metastases without other involvement at the time of SM resection. While a few cases were reported where a resection for another metastatic site was performed before or after the splenectomy for SM, these cases were not excluded.

### 2.2. Statistical Analysis

Numeric data are presented as median (range), while categorical variables are expressed as numbers (percentages). However, the survival is presented as median (range) and mean (±SD) because the median was not reached in several survival analyses due to the high number of censored patients. Fisher’s exact test (two-tailed) was used to compare the categorical variables between the groups. The survival curves were estimated using the Kaplan–Meier method and were compared using the long-rank test. The median follow-up time was assessed using the reversed Kaplan–Meier curves. A *p*-value < 0.05 was considered statistically significant.

## 3. Results

The search in the abovementioned databases identified 83 patients with splenectomies for metachronous isolated SM of CRC origin (no other sites of metastases at the time of splenectomy for SM), extracted from papers published between 1965 and 2024 (Appendix A, references [16,17,18,20,25,26,27,28,29,30,31,32,33,34,35,36,37,38,39,40,41,42,43,44,45,46,47,48,49,50,51,52,53,54,55,56,57,58,59,60,61,62,63,64,65,66,67,68,69,70,71,72,73,74,75,76,77] are cited in Appendix A). The median age of patients at the time of splenectomy was 62 years (range, 22–84 years), with a slightly male gender predominance (50.6%).

The primary CRC origin was the left colon (including the sigmoid) in 38 patients (45.8%), the right colon in 24 patients (28.9%), and the rectum in 10 patients (12%). Two patients (2.4%) had the transverse colon as the primary site of the tumor, and one patient (1.2%) had a double primary tumor location (left and right colon). In eight patients (9.6%), the primary tumor location was the colon, but the exact location was not specified. Thus, overall, 73 patients (88%) had the colon as the primary tumor location.

Data about T, N, and Duke’s stages of the primary CRC were available for 44 patients (53%), 45 patients (54.2%), and 65 patients (78.3%), respectively. Thus, 28 patients (out of 44 patients—63.6%) have had T3 stages, 12 patients (27.3%) have had T4 stages, and only 4 patients (9.1%) have had a T2 stage. No patient was in a T1 stage. Loco-regional positive lymph nodes of the resected primary CRC were observed in 27 patients (out of 45 patients—60%). Regarding the Duke’s stage of the primary CRC, the most significant part of the patients were Duke C—45 patients (out of 64 patients—70.3%), followed by Duke’s B stage—13 patients (20.3%); Duke’s A and D stages were observed in three patients in each group (13.7%). Negative resection margins were obtained in all patients with available data after primary CRC resection (available data in 45 patients—54.2%).

An adjuvant chemotherapy after resection of the primary CRC was performed in 40 patients (out of the 52 patients with available data)—76.9%. The most frequently used chemotherapy regimens included 5-fluorouracil (10 patients out of 40 patients with chemotherapy—25%), FOLFOX (8 patients—20%), and capecitabine (3 patients—7.5%); in the remaining patients with adjuvant chemotherapy (18 patients—47.5%), the used regimen was not specified.

The median interval between primary CRC resection and the splenectomy for SM was 24 months (range, 3–180 months) (data available for 79 patients—95.2%).

Data about clinical signs and symptoms, imaging, and CEA serum level at the time of splenectomy for SM were available for 79 patients (95.2%), 74 patients (89.2%), and 63 patients (75.9%), respectively.

The most significant portion of patients were asymptomatic at the time of splenectomy for SM—66 patients (out of 79 patients with available data—83.5%). Only 13 patients were symptomatic (16.5%). The main clinical signs and symptoms are shown in Table 1.

The imaging methods for detecting metachronous isolated SM of CRC are shown in Table 2. Thus, a contrast-enhanced computed tomography (CT) was performed in the most significant portion of patients—64 patients (out of 74 patients—86.5%), followed by a positron emission computed tomography (PET-CT) (22 patients—29.7%).

An elevated CEA serum level was observed at the time of splenectomy for SM of CRC origin in 52 patients (out of 63 patients with available data—82.5%). The median CEA serum level in the present cohort of patients was 26.2 ng/mL (range, 2.6–363 ng/mL) (data available for 51 patients—61.4%).

The most significant portion of the patients in the present cohort underwent an open splenectomy (68 patients—81.9%), while the remaining 15 patients (18.1%) were laparoscopically approached. In the open approach group of patients, two patients (2.4%) have had associated procedures (right oophorectomy for a serous cyst in one patient and partial gastrectomy in the other patient). However, in the last 20 years (2003–2024), the rate of laparoscopic splenectomies for SM of CRC origin was significantly higher than the rate reported between 1965 and 2002 (33.3% vs. 0%, *p* < 0.001).

Data about the number and size of SM of CRC were available in 75 patients (90.4%) and 63 patients (75.9%), respectively. Thus, a solitary metastasis was observed in 69 patients (out of 75 patients with available data—92%), two metastases in 4 patients (5.3%), and multiple spleen metastases in 2 patients (2.7%). The median diameter of the SM was 4.5 cm (range, 1–18 cm).

Data about perioperative chemotherapy at the time of splenectomy for SM of CRC origin were available in 47 patients (56.6%). Thus, chemotherapy was performed in 22 patients (out of 47 patients with available data—46.8%). The chemotherapy regimens are shown in Table 3. Targeted therapy was used in only six patients (27.3%). It is worth mentioning that in the last 20 years (2003–2024), the rate of perioperative chemotherapy at the time of splenectomy for SM of CRC origin was significantly higher than the rate reported between 1965 and 2002 (60% vs. 8.3%, *p* = 0.002).

Data about the overall survival and status were available in 70 patients (84.3%). Thus, 61 patients (out of 70 patients with available data—87.1%) were alive at the time of the last follow-up (53 patients without recurrence—75.7% and 8 patients with disease recurrence—11.4%). Only nine patients (12.6%) died of the disease recurrence during the follow-up after splenectomy for SM of CRC. The estimated median overall survival time after splenectomy was 84 months (range, 1–87 months; mean 66.5 ± 5.9 months). However, the median follow-up time after splenectomy was only 12 months (range, 1–87 months; mean 22.2 ± 2.8 months).

The estimated one-year and five-year overall survivals in the present cohort were 96% and 78%, respectively (Figure 1).

Furthermore, several comparative survival analyses were performed, based on the available data in each group, to assess potential predictors of long-term survival in patients with splenectomies for metachronous isolated SM of CRC origin (Table 4). Thus, no significant impact on the long-term outcomes was observed for factors such as gender, primary tumor location (colon vs. rectum, left vs. right colon), Duke’s stage of the primary tumor, presence or absence of positive lymph nodes after primary tumor resection, presence or absence of adjuvant chemotherapy after primary tumor resection or after SM resection, interval time from primary tumor resection to splenectomy for SM, presence or absence of symptoms or signs, the value of CEA serum level, type of approach for splenectomy (laparoscopic vs. open), number and size of SM, and period of time (1965–2002 vs. 2003–2024) (*p* values ≥ 0.070, ns). Interestingly, patients younger than 62 presented statistically significantly worse overall survival rates than those ≥62 years old (*p* = 0.011), as shown in Table 4 and Figure 2.

## 4. Discussion

Intraparenchimatous SM from solid tumors is widely considered exceptional and usually occurs in the context of multi-visceral disseminated cancer (more than 50% of the cases) and rarely as a solitary lesion. Autopsy, ultrasonography, and splenectomy series have reported a broad spectrum of prevalence for SM in solid cancers, ranging between 0.15% and 9.8% [24,33,37,51,78,79,80,81,82]. In the context of multi-visceral metastatic disease, the most common sources of SM are melanoma (34%), breast cancer (12%), and ovarian cancer (12%), with only 10% arising from CRC [71]. Other studies identified lung cancer (18.6–30.5%), melanoma (8.4–15.8%), and breast cancers (10.2–12.3%) as the most frequent sources of SM [24,79,83]. A review published in 2001 identified 50 patients with splenectomies for SM, gynecological cancers being the most frequent source (60%), while CRC origin was observed in 11% of the patients [84]. Another study, including 93 patients with solitary SM, identified CRC and ovarian cancer as the most frequent sources [80]. Nevertheless, a few extensive studies analyzing the data of 84 to 115 patients with splenectomies for SM identified as the most frequent primary sources of SM, ovarian cancer (31.3–46.4%), melanoma (16.7–27.8%), and CRC (10.7–13.9%) [56,85].

An autopsy study published in 1974 reported a 4.4% incidence of SM from CRC origin (none as isolated SM) [86], while a few clinical studies identified an incidence of 0.2% to 0.5% for isolated SM from CRC origin [24,37,60].

Several studies reviewing the patients with splenectomies for SM of CRC origin reached conflicting results about the number of published cases [16,17,18]. In a study published in 2007, 42 patients with isolated SM of CRC origin were described in the literature [19], while another study published in 2016 identified 48 patients with SM of CRC origin [20]. Interestingly, in a study published in 2022, only 39 patients with isolated SM of CRC were retrieved from the English literature [22]. Nevertheless, a Japanese-language paper analyzed the data of 75 Japanese patients with synchronous and metachronous SM of CRC origin [21]. To the best of our knowledge, the present study is the largest to date, including only patients with splenectomies for metachronous isolated SM of CRC origin.

The presumed factors that may restrict metastasis to the spleen include the constant high blood flow through the spleen with rhythmic contraction by the sinusoidal splenic architecture (not allowing potential tumor embolus to soil), the sharp angle of the splenic artery with the celiac axis, the absence of afferent lymphatics to the spleen, the scarcity of lymphatic vessels extending into the intrasplenic parenchyma, the splenic capsule (acting as a shield that prevents intraparenchymal metastasis), and the good immune surveillance in the spleen that inhibits tumor cell proliferation [16,44,51,87]. Old experimental studies have shown that the growth rate of inoculated adenocarcinoma cells into the spleen is significantly slower compared with the liver [25]. Conversely, the propensity of CRC to spread using the lymphatic and vascular systems may explain, at least in part, the high incidence of liver, lung, and lymph node metastases [4]. A hypothesis suggested that the rarity of SM in epithelial cancers originates from cancer cells undergoing pooling within the spleen, being exposed to pro-apoptotic signals, and consequently failing to survive [88]. Nevertheless, the rarity of metastases of solid cancers into the spleen remains unexplained mainly, even though it is the most vascular organ.

Several reviews reporting patients with splenectomies for SM of CRC origin, identified the colon (88–92.3%), particularly the left colon, as the most frequent site for the primary tumor (61.5–88%), presenting vastly advanced stages of the primary tumor (Duke’s stages C in 63–67.9% of the patients), with a median interval time from primary tumor resection to SM diagnosis/resection of 18 months (range, 1–144 months), the most significant portion of patients being asymptomatic at the time of SM diagnosis (78.6%). An elevated CEA serum level was observed in 55% to 83.9% of the patients at the time of SM diagnosis, while the imaging diagnosis included CT scan in 87.1% of the patients and PET-CT in 12.9% of the patients. A metachronous pattern for SM was reported in 83.9–92.9% of the patients, the most significant portion of SM being solitary (92–99%) and treated with splenectomy (94–96.4%) [17,18,19,20,22,65,87].

In the present cohort, the colon was the most frequent site for the primary tumor (88%), with the left colon as the first source (45.8%). Advanced disease of the primary tumor (Duke’s C) was observed in 70.3% of the patients, with a median interval from primary tumor resection to splenectomy for SM of CRC origin of 24 months (range, 3–180 months). A solitary SM was observed in 92% of the patients. The characteristics of the patients in our study appear consistent with those reported in previous studies addressing the same topic [17,18,19,20,22,65,87]. Interestingly, the study of Japanese patients by Kurumiya and co-workers in 2019 identified the right colon as the leading site for primary tumors (38.7%) [21].

Except for the study of Kurumiya and co-workers [21], all other studies (including the present one) identified the left colon as the primary source for SM of CRC origin, as highlighted above. One might explain this feature by the possibility of a retrograde spread from the inferior mesenteric vein to the spleen via the splenic vein [34], particularly in portal hypertension [31]. The prevalence of the right colon as a primary source of SM of CRC origin in the Japanese study of Kurumiya and co-workers [21] appears to be at odds with the theory of Indudhara and co-workers for SM occurrence in CRC [34]. Different metastatic patterns were previously described between the colon and rectal cancers [5,7]. A previous study has shown that the prevalence of liver, lung, and bone metastases is higher in the left vs. right colon cancer [5]. Rectal cancers spread significantly more frequently to the thorax, the bone, and the nervous system than colon cancers [7].

It is widely accepted that patients with metastatic disease after curative intent therapy for CRC usually will develop the metastases within the first three years after surgery (86%), with a median time to diagnosis of first metastasis of 17 months (range, 10–29 months). The median time to diagnose liver metastases is 15 months, while for lung and peritoneum metastases, the median time to diagnosis is 22 months and 16 months, respectively [4,7]. In the present cohort, the median time from primary CRC resection to SM resection was 24 months (range, 3–180 months). Furthermore, in a study of 93 patients with solitary SM of different types of cancer origin, the median interval time from the primary tumor resection to the SM diagnosis was 28 months (range, 0–264 months); for SM of CRC origin, the median interval time from the primary tumor to the SM diagnosis was 27 months (range, 0–132 months) [80]. Thus, one might speculate that metachronous SM occurs later after CRC resection, compared with the liver, lung, or peritoneal metastases. The slow progression by the splenic function of immune surveillance might explain the long interval between resection of the primary CRC and SM occurrence [16].

Most patients in the present cohort were asymptomatic at the time of SM diagnosis (83.5%), consistent with previously reported series of SM of CRC origin [17,18,19,20,22,65,87]. In a limited number of patients reported in the literature, SM may be complicated by splenic and portal vein thrombosis [89], spleen abscess [19], or spleen rupture [36,90].

An elevated CEA serum level was present in a high proportion of patients in the present cohort (82.5%), leading to further imaging investigations to detect SM. In almost all cases of the present study, radiologic diagnosis was established by CT (performed in 86.5% of the patients), followed by PET-CT (performed in 29.7%). It is worth mentioning the case of two patients in whom SM was first diagnosed on the operative specimen of splenectomy, and further investigations led to a colon carcinoma diagnosis [90,91].

Whenever a splenic mass is detected by imaging (particularly in the context of a patient with previous malignancies, including CRC), it raises the suspicion of metastatic disease based on clinical history, CEA serum levels, and appearance on imaging studies, and it is also necessary to exclude the presence of other metastatic lesions by a PET-CT. A previous history of cancer has been identified as the single independent predictor of malignancy in a splenic lesion [85].

SM should be differentiated from other primary lesions in the spleen, particularly in the context of liberal use and development of modern imaging techniques. Of note, the most common tumors of the spleen are benign; isolated metastatic lesions to the spleen remain an exceptional appearance. Other lesions, such as hemangioma, infarction, and spleen infectious or inflammatory conditions, should also be ruled out. Thus, a new lesion in the spleen in the imaging follow-up of a patient who underwent resection for CRC may pose some challenging problems of diagnosis and treatment. An accurate imaging differentiation of benign/malignant spleen lesions is critical for the proper management, particularly in the context of a patient with previous malignancies; commonly, an SM is hypoechoic on ultrasonography, hypodense on contrast-enhanced CT in the venous phase, hypointense on T1-weighted MRI sequences images following contrast administration, and intense hypermetabolic on PET-CT [58,92,93,94,95,96]. It is worth mentioning the case of a patient diagnosed with concurrent rectosigmoid carcinoma and primary splenic malignant lymphoma mimicking an SM of CRC origin [97]. CT scans reveal the most significant part of SM. However, a PET-CT is recommended not necessarily to identify the SM but to rule out other metastatic sites. Mestner and co-workers showed a 100% sensitivity and specificity of PET-CT in diagnosing SM [94].

Although in the past, the presence of SM in solid cancers was considered an advanced disease with no indication for surgery [98], nowadays, for isolated SM, total splenectomy is the treatment of choice. This situation is associated with encouraging survival [24,87]. While an open technique in the setting of previously operated patients seems the first choice [87], other studies suggest that a laparoscopic approach can be successfully performed for isolated SM of solid cancers, with all the advantages of minimally invasive surgery [60,69,99,100,101]. The oncological safety of a minimally invasive approach for spleen malignancies (including SM) was recently demonstrated [102]. A minimally invasive splenectomy is associated with low morbidity rates and almost nil mortality rates for both benign and malignant pathologies [102,103], although a malignant pathology is a risk factor for postoperative morbidity [104].

In the present cohort, SM was predominately resected by open splenectomy (81.9%). However, in the last 20 years (2003–2024), the rate of laparoscopic splenectomies for SM of CRC origin was significantly higher than the rate reported between 1965 and 2002 (33.3% vs. 0%, *p* < 0.001). One might conclude that the laparoscopic approach will be increasingly used as a standard of care for the few patients diagnosed with SM of solid cancers, including CRC.

While not the particular subject of this paper, there are reports of partial splenectomy being performed for SM [105,106,107,108], and thus, this may be a theme for future research. Partial splenectomy, including the laparoscopic approach, has been demonstrated as safe in experienced surgeon hands, with low morbidity rates, being proposed as an alternative to total splenectomy for a specific pathology to overcome potential complications of asplenia; a malignant pathology is, however, a very rare indication for partial splenectomy (1.5%) [106,107,109,110]. Indeed, total splenectomy may induce several significant complications: infections (including severe ones and mortality related to infections, which is 2–3-fold greater than that of the general population), vascular complications (such as venous thromboembolism, stroke, myocardial infarction, and pulmonary hypertension), and a high risk of developing cancer [111,112]. A total splenectomy may induce immunodeficiency, including a decrease in pro-inflammatory cytokines; this change is hindered with a partial splenectomy, particularly with a minimally invasive approach [113]. A few studies have associated total splenectomy with worse survival rates in CRC [114,115], while other studies did not find any significant impact [116]. Furthermore, one experimental study has shown that the growth rate of hematogenous lung metastases decreases after splenectomy [117], while another study has associated splenectomy with increased liver metastases in CRC [118].

For the exceptional cases of synchronous SM of CRC origin, resection of both sites in the same surgical session is recommended; a sequential approach including first resection of the primary tumor and splenectomy at a later time might be a safe option for patients with high surgical risks [119].

The liver is the most common site of metastasis from CRC (up to 50% of the patients), followed by the lungs (10–15% of the patients) [120]. The median overall survival of patients resected for metastatic CRC is reported to be around 38–59 months for liver-only metastases and 45–64 months for lung-only metastases [6,11,121,122,123]. Resection for peritoneal metastases of CRC origin is associated with a median overall survival time of 47.7 months [15]. Resection for isolated lung metastasis of CRC origin is associated with five- and ten-year survival rates of 25–58% and 17–28%, respectively. Resection for isolated liver metastasis of CRC origin is associated with five- and ten-year survival rates of 25–58% and 17–30%, respectively [120,121,122,123,124,125].

A study has shown that SM is associated with a worse prognosis than other metastatic sites in solid cancers [79], a situation that appears to be at odds with the results of the present study. In the present cohort, the median overall survival after splenectomy for isolated SM of CRC origin was 84 months, with 96% one-year and 78% five-year survival rates. Similar long-term outcomes were previously reported [17,18,19,20,22,65,87]. Thus, one might speculate that splenectomies for isolated SM from CRC origin are associated with encouraging long-term survival, apparently better than those reported after resectioning other isolated metastatic sites such as the liver or lung.

The present study did not find any significant impact on the long-term outcomes for factors such as gender, primary tumor location (colon vs. rectum, left vs. right colon), Duke’s stage of the primary tumor, presence or absence of positive lymph nodes after primary tumor resection, presence or absence of adjuvant chemotherapy after primary tumor resection or after SM resection, interval time from primary tumor resection to splenectomy for SM, presence or absence of symptoms or signs, the value of CEA serum level, type of approach for splenectomy (laparoscopic vs. open), number and size of SM, and period of time (1965–2002 vs. 2003–2024) (*p* values ≥ 0.070, ns).

Only one study in the English literature assesses potential prognostic factors for long-term survival in patients with splenectomies for SM of CRC origin showing no impact on survival for the age, gender, primary tumor location, Duke’s stage, SM size, synchronous/metachronous pattern, or CEA serum level in this study [17]. Another study reflecting the data of 75 Japanese patients with splenectomies for synchronous and metachronous SM of CRC origin (1989–2018) did not identify any impact on the long-term survival for the primary tumor site (right vs. left colon) or synchronous vs. metachronous pattern [21].

A study of 115 patients with splenectomies for SM of different solid cancer origin did not find any influence of age at time of splenectomy, gender, and the primary cancer origin on the long-term survival [56]; however, it appears that metachronous SM has statistically significant better long-term outcomes compared with synchronous SM [24,56].

Similar results (as in the studies mentioned above) were obtained in the present study, except for age. Thus, in the present study, patients younger than 62 years presented statistically significantly worse overall survival rates after splenectomies for SM of CRC origin than those ≥62 years (*p* = 0.011), as shown in Table 4 and Figure 2. Younger ages were previously associated with worse survival also after hepatectomies for liver metastases of CRC origin in a study [126]. Conversely, another study associated age < 65 years with better overall survival after resectioning liver and lung metastases of CRC origin [127].

A few studies have shown that the primary tumor site (left vs. right colon) appears to influence long-term outcomes after resection of liver and lung metastases of CRC origin but reached conflicting results of which location has the worst survival [127,128,129]. Other studies did not identify any prognostic value of the primary tumor site on the long-term outcomes after resection for liver or lung metastases of CRC origin [121,130], as was the case in the present study for SM of CRC origin.

No influence of the preoperative CEA serum level on the long-term outcomes after splenectomies for SM of CRC origin was reported in the present study. This situation is at odds with previous studies showing a prognostic role for preoperative CEA serum level after resection for liver and lung metastases of CRC origin [124,131].

In two studies, the overall survival was significantly better for patients resected for isolated SM of CRC origin compared with those with associated other metastatic sites at the time of splenectomy for SM [17,21]. A study on 26,170 patients with stage IV CRC identified the distant metastasis site and the number of metastasis sites as independent prognostic survival factors, highlighting the diverse treatment strategies for patients with different metastatic patterns [9].

Kurumiya and co-workers have shown that the five-year survival rates were significantly better for patients in the interval 2003–2018 (introduction of oxaliplatin in the adjuvant chemotherapy after splenectomy for SM) vs. 1989–2002 (80.8% vs. 50.1%, *p* = 0.031) [21]. Our study failed to show any significant improvement in survival in the last 20 years (Table 4). This is particularly interesting in the context of the fact that in the last 20 years (2003–2024), the rate of perioperative chemotherapy at the time of splenectomy for SM of CRC origin was significantly higher than the rate reported between 1965 and 2002 (60% vs. 8.3%, *p* = 0.002) in the present study. However, our analyses did not associate adjuvant chemotherapy with improved survival (Table 4). Another explanation for the different survival analyses in the two studies might be related to the differences regarding the rate of perioperative chemotherapy at the time of SM resection (87.5% in the Japanese study [21] and only 46.8% in our study). Furthermore, in our study, there was considerable heterogeneity regarding the chemotherapy protocols, as shown in Table 4.

The present study has a few significant limitations: the number of analyzed patients is low because SM of CRC origin is an exceptional pathology; the heterogeneity of the literature data may impact the results of the analyses performed in the present study, particularly regarding those related to oncological medical therapy and long-term survival; the short follow-up time after splenectomy for SM of CRC represents another limitation of the study, along with the high number of censored patients. Thus, the results of the present study should be regarded with caution for clinical decision-making, considering that excellent survival rates are probably overestimated.

## 5. Conclusions

Our literature search shows that the spleen remains an exceptional site for metastatic disease in CRC. SM appears to develop later after primary CRC resection, compared with other metastatic sites such as the liver, lungs, and peritoneum. An accurate imaging differentiation of benign/malignant spleen lesions is critical for proper management, particularly in a patient with previous malignancies. Splenectomy is the optimal treatment for patients with isolated metachronous SM of CRC origin fit for surgery, with the laparoscopic approach being increasingly used and having the potential to become a standard of care for the few patients diagnosed with SM of solid cancers, including CRC. Encouraging long-term survival rates were reported after splenectomies for isolated metachronous SM of CRC in the context of a multidisciplinary approach. Perioperative chemotherapy in the context of a patient diagnosed with SM of CRC origin appears to be a reasonable option, although the present study failed to show any significant impact on long-term survival. Younger ages are associated with worse survival. Further studies should focus on the potential benefit of partial splenectomies and modern chemotherapy regimens for SM of solid organ cancers, including CRC.

## Figures and Tables

**Figure 1 jcm-13-02362-f001:**
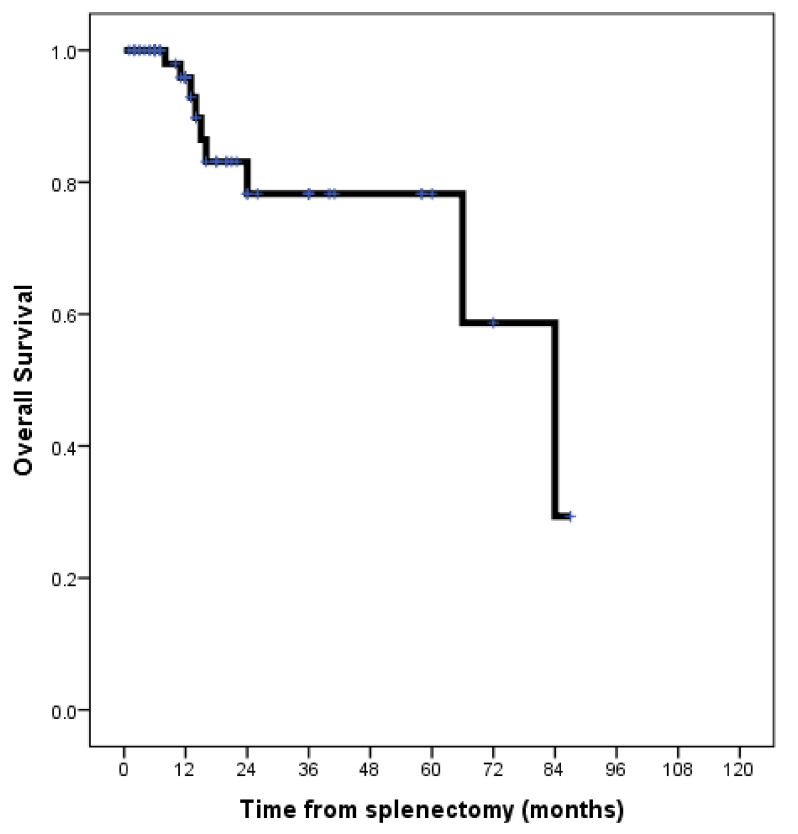
Kaplan–Meier survival curve estimating the overall survival in 70 patients with splenectomies for splenic metastases of colorectal cancer origin.

**Figure 2 jcm-13-02362-f002:**
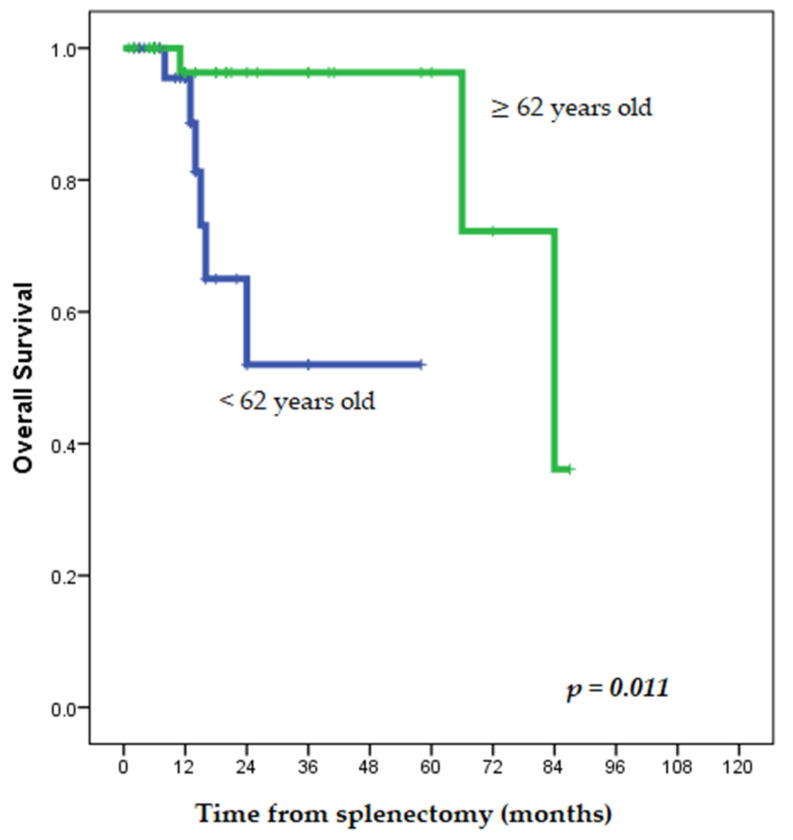
Kaplan–Meier comparative survival curves for the overall survival in the group of patients < 62 years old vs. ≥ 62 years old with splenectomies for splenic metastases of colorectal cancer origin.

**Table 1 jcm-13-02362-t001:** Clinical signs and symptoms in 79 patients with splenectomies for metachronous isolated SM of CRC origin.

Clinical Signs and Symptoms	No of Patients (%)
Asymptomatic	66 patients (83.5%)
Symptomatic:	13 patients (16.5%)
Abdominal pain	7 patients (8.9%)
Fatigue	1 patient (1.3%)
Fever	1 patient (1.3%)
Hematuria	1 patient (1.3%)
Malaise	1 patient (1.3%)
Weight loss and abdominal pain	1 patient (1.3%)
Hemoperitoneum due to rupture of the spleen	1 patient (1.3%)

SM—splenic metastasis; CRC—colorectal cancer.

**Table 2 jcm-13-02362-t002:** Imaging methods for diagnosis in 74 patients with splenectomies for metachronous isolated SM of CRC origin.

Imaging Method	No of Patients (%)
CT scan only	45 patients (60.8%)
CT and PET-CT	15 patients (20.3%)
PET-CT only	6 patients (8.1%)
CT and MRI	2 patients (2.7%)
Ultrasonography only	2 patients (2.7%)
Radionuclide-only liver-spleen scintigraphy	1 patient (1.4%)
CT and radionuclide liver-spleen scintigraphy	1 patient (1.4%)
CT, PET-CT, and PET-MRI	1 patient (1.4%)
CT and radionuclide liver-spleen scintigraphy	1 patient (1.4%)
Abdominal-only roentgenography	1 patient (1.4%)

SM—splenic metastasis; CRC—colorectal cancer; CT—computed tomography; PET-CT—positron emission tomography; MRI—magnetic resonance imaging; PET-MRI—positron emission magnetic resonance imaging.

**Table 3 jcm-13-02362-t003:** Chemotherapy regimens used in 22 patients at the time of splenectomies for metachronous isolated SM of CRC origin.

Chemotherapy Regimen	No of Patients (%)
XELOX	3 patients (13.6%)
FOLFIRI and targeted therapy	3 patients (13.6%)
5-fluorouracil	2 patients (9.1%)
FOLFOX	2 patient (9.1%)
FOLFOX and targeted therapy	1 patient (4.5%)
5-fluorouracil and targeted therapy	1 patient (4.5%)
Capecitabine	1 patient (4.5%)
Targeted-only therapy	1 patient (4.5%)
Not specified	8 patients (36.6%)

SM—splenic metastasis; CRC—colorectal cancer.

**Table 4 jcm-13-02362-t004:** Exploring potential predictors for the overall survival in patients with splenectomies for metachronous isolated SM of CRC origin.

Parameter	Median OS, Months	Mean OS, Months	1-Year OS, %	5-Year OS, %	*p* Value
Gender					0.880, ns
M	84 (1–84)	69.8 ± 9	95%	78%	
F	NR (2–87)	64.7 ± 8.1	95%	77%	
Age					0.011
<62 years	NR (2–58)	38 ± 6.2	96%	52%	
≥62 years	84 (1–87)	78 ± 4.9	96%	96%	
Primary tumor					0.571, ns
Colon	66 (1–87)	63.4 ± 7.1	95%	74%	
Rectum	84 (3–84)	84	100%	100%	
Primary tumor					0.689, ns
Right colon	NR (2–87)	65.7 ± 9.1	94%	71%	
Left colon (including sigmoid)	66 (1–66)	55.6 ± 9.1	96%	76%	
Duke’s stage					0.616, ns
A-B	66 (2–66)	61.4 ± 6.2	91%	91%	
C-D	84 (1–87)	74.7 ± 5.9	96%	85%	
N stage					0.157, ns
Negative	66 (2–66)	66	100%	100%	
Positive	84 (1–87)	85.5 ± 1.1	100%	100%	
Adjuvant chemotherapy after primary CRC resection					0.473, ns
Yes	NR (1–87)	80 ± 6.6	100%	84%	
No	66 (12–84)	75.7 ± 9	100%	100%	
Interval from primary tumor resection					0.070, ns
<24 months	NR (2–58)	39.3 ± 6.6	95%	54%	
≥24 months	84 (1–87)	72.4 ± 6.6	95%	90%	
Signs or symptoms					0.356, ns
Yes	84 (4–84)	65.4 ± 14	75%	75%	
No	66 (1–87)	65.3 ± 7.9	100%	81%	
CEA serum level					0.122, ns
<26.2 ng/mL	NR (6–60)	55.8 ± 4	100%	91%	
≥26.2 ng/mL	66 (2–87)	58.7 ± 10.4	83%	73%	
Splenectomy approach					0.414, ns
Open	NR (2–87)	NA	95%	76%	
Laparoscopic	NR (1–36)	NA	100%	100%	
Number of SM					0.597, ns
1	NR (1–87)	NA	95%	81%	
≥2	NR (5–12)	NA	100%	100%	
Diameter of SM					0.689, ns
<4.5 cm	66 (3–87)	66.1 ± 8.3	95%	82%	
≥4.5 cm	84 (1–84)	71.3 ± 10.5	94%	82%	
Adjuvant chemotherapy after SM resection					0.141, ns
Yes	NR (1–58)	48.7 ± 5.8	100%	75%	
No	84 (5–87)	79 ± 5.4	100%	100%	
Period					0.105, ns
1965–2002	84 (2–84)	62.2 ± 8.7	95%	75%	
2003–2024	NR (1–87)	77.6 ± 6.2	100%	87%	

SM—splenic metastasis; CRC—colorectal cancer; OS—overall survival; CEA—carcinoembryonic antigen; NR—not reached due to the increased number of censored patients; NA—not available; ns—not significant.

## Data Availability

Data are provided in Appendix A.

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
