# Peer review of "A Glimpse into the Role and Effectiveness of Splenectomy for Isolated Metachronous Spleen Metastasis of Colorectal Cancer Origin: Long-Term Survivals Can Be Achieved"

_jcm, 2024, doi:10.3390/jcm13082362_

Round 1

Reviewer 1 Report

Comments and Suggestions for Authors

This article provides a compelling and insightful analysis of the role and effectiveness of splenectomy in patients with isolated metachronous spleen metastasis from colorectal cancer, suggesting that long-term survival can be achieved with this approach. The authors have done commendable work in collecting and analyzing data to support their hypothesis. The methodology is sound, and the statistical analysis adds robustness to the findings. Moreover, the literature review is comprehensive, and the discussion provides a thoughtful exploration of the potential mechanisms and clinical implications of the study findings.

However, there are a few areas that could benefit from further refinement:

The Results section:

1 The text does not clearly outline the statistical methods used for analyzing the data, particularly in comparative survival analyses. A more detailed description of the statistical tests, confidence intervals, and the criteria for statistical significance (e.g., p-value < 0.05) would strengthen the credibility and interpretability of the results.

2 Some sections of the results could benefit from more concise language without sacrificing completeness. For instance, the detailed enumeration of chemotherapy regimens and their percentages might be more effectively presented in a table, which would make the text more readable and allow for easier comparison.

3 In regards to the division of left and right halves of colorectal cancer, the author's method of division seems to diverge from the current mainstream approaches. What is the basis for the author's method of division?

The Patients and Methods section:

1 The methods section does not mention inter-rater agreement verification among multiple researchers during the literature screening and data extraction process. Considering that this process could impact the final study results and conclusions, it is suggested that the authors include such information in the methods section.

2 While exclusion criteria are mentioned, it is not explicitly stated how missing or incomplete clinical data were handled. When describing the data collection and analysis process, it is recommended that the authors briefly outline the methods used to address such issues.

Comments on the Quality of English Language

The language used is clear, and the technical terminology is appropriate for the intended academic audience.

Author Response

Response to Reviewer 1

  1. This article provides a compelling and insightful analysis of the role and effectiveness of splenectomy in patients with isolated metachronous spleen metastasis from colorectal cancer, suggesting that long-term survival can be achieved with this approach. The authors have done commendable work in collecting and analyzing data to support their hypothesis. The methodology is sound, and the statistical analysis adds robustness to the findings. Moreover, the literature review is comprehensive, and the discussion provides a thoughtful exploration of the potential mechanisms and clinical implications of the study findings.

Response 1: Thank you for your kind comments and appreciation.

 However, there are a few areas that could benefit from further refinement:

 The Results section:

  1. The text does not clearly outline the statistical methods used for analyzing the data, particularly in comparative survival analyses. A more detailed description of the statistical tests, confidence intervals, and the criteria for statistical significance (e.g., p-value < 0.05) would strengthen the credibility and interpretability of the results.

Response 2: We described the statistical tests and statistical significance in Patients and Methods, Statistical analysis.

  1. Some sections of the results could benefit from more concise language without sacrificing completeness. For instance, the detailed enumeration of chemotherapy regimens and their percentages might be more effectively presented in a table, which would make the text more readable and allow for easier comparison.

Response 3: The chemotherapy regimens and their percentages are presented in Table 4.

  1. In regards to the division of left and right halves of colorectal cancer, the author's method of division seems to diverge from the current mainstream approaches. What is the basis for the author's method of division?

Response 4: For each of the 83 cases retrieved from the literature, we extracted the right and left colons as they were provided in the searched literature. We just considered the left colon for both descending and sigmoid cancers because it has the same venous drainage through the inferior mesenteric vein. Some authors suggest that this is why left colon cancer more frequently metastasizes in the spleen. However, other studies did not confirm this theory (all these issues are discussed in the present manuscript).

 The Patients and Methods section:

  1. The methods section does not mention inter-rater agreement verification among multiple researchers during the literature screening and data extraction process. Considering that this process could impact the final study results and conclusions, it is suggested that the authors include such information in the methods section.

Response 5: We have introduced a paragraph addressing the suggested issue (lines 116 – 118). The track changes function highlighted the modifications to the manuscript.

  1. While exclusion criteria are mentioned, it is not explicitly stated how missing or incomplete clinical data were handled. When describing the data collection and analysis process, it is recommended that the authors briefly outline the methods used to address such issues.

Response 6: In the Results, the number of patients with available data for analyses was mentioned at each point for a few parameters that were not available for all patients. Furthermore, the heterogeneity of such data was mentioned as a limitation of this study.

Reviewer 2 Report

Comments and Suggestions for Authors

Dear Authors

Systematic review describes well about “Splenic Metastases (SM) appears to develop later after primary CRC resection, compared with other metastatic sites such as the liver, lungs, and peritoneum. Most of the patients with SM are asymptomatic at diagnosis; spleen rupture due to SM is exceptional. An elevated CEA serum level during the follow-up after primary CRC resection raised awareness of a potential disease recurrence and led to further imaging exploration to detect SM”.

Perioperative chemotherapy in the context of a patient diagnosed with SM of CRC origin appears to be a reasonable option, albeit the present study failed to show any significant impact on long-term survival.

Overall, the review is written well.

Author Response

  1. Systematic review describes well about “Splenic Metastases (SM) appears to develop later after primary CRC resection, compared with other metastatic sites such as the liver, lungs, and peritoneum. Most of the patients with SM are asymptomatic at diagnosis; spleen rupture due to SM is exceptional. An elevated CEA serum level during the follow-up after primary CRC resection raised awareness of a potential disease recurrence and led to further imaging exploration to detect SM”. Perioperative chemotherapy in the context of a patient diagnosed with SM of CRC origin appears to be a reasonable option, albeit the present study failed to show any significant impact on long-term survival. Overall, the review is written well.

Response 1: Thank you for your kind comments and appreciation.

Reviewer 3 Report

Comments and Suggestions for Authors

Isolated metachronous spleen metastasis of colorectal cancer origin is a rare occurrence, and the role of splenectomy in managing such cases has been a subject of debate. However, some studies suggest that splenectomy can be effective in achieving long-term survival in select patients.

The rationale behind splenectomy in this context lies in the potential for achieving complete resection of metastatic disease, which is often associated with better outcomes compared to unresectable metastases. Additionally, the spleen is considered an uncommon site for metastasis from colorectal cancer, and isolated metastases to the spleen without involvement of other organs may indicate a more favorable disease biology.

The topic is undeniably captivating and pertinent. It focuses on a particular clinical situation that is relatively rare but important in the treatment of individuals with colorectal cancer. Investigating the function of splenectomy in these instances might provide significant knowledge on treatment approaches and results for this specific group of patients. Moreover, comprehending the efficacy of splenectomy in attaining prolonged survivorship may have ramifications for clinical decision-making and the provision of patient care. In general, this topic is captivating and has the potential to enhance the current understanding in the fields of cancer and surgery.

Author Response

  1. Isolated metachronous spleen metastasis of colorectal cancer origin is a rare occurrence, and the role of splenectomy in managing such cases has been a subject of debate. However, some studies suggest that splenectomy can be effective in achieving long-term survival in select patients.The rationale behind splenectomy in this context lies in the potential for achieving complete resection of metastatic disease, which is often associated with better outcomes compared to unresectable metastases. Additionally, the spleen is considered an uncommon site for metastasis from colorectal cancer, and isolated metastases to the spleen without involvement of other organs may indicate a more favorable disease biology.The topic is undeniably captivating and pertinent. It focuses on a particular clinical situation that is relatively rare but important in the treatment of individuals with colorectal cancer. Investigating the function of splenectomy in these instances might provide significant knowledge on treatment approaches and results for this specific group of patients. Moreover, comprehending the efficacy of splenectomy in attaining prolonged survivorship may have ramifications for clinical decision-making and the provision of patient care. In general, this topic is captivating and has the potential to enhance the current understanding in the fields of cancer and surgery.

Response 1: Thank you for your kind comments and appreciation.